# The relationship between smoking cigarettes and metabolic syndrome: A cross-sectional study with non-single residents of Seoul under 40 years old

**Sun Woo Kim**[1⊙], **Ho Jun Kim**[2⊙], **Kyungha Min**[2], **Hobeom Lee**[2], **Sung-Ha Lee**[3], **Sunyoung Kim**[4], **Jong Seung Kim**[2], **Bumjo Oh**[2]*

1 Department of Family Medicine, Bumin Hospital, Seoul, Republic of Korea, 2 Department of Family Medicine, SMG-SNU Boramae Medical Center, Seoul, Republic of Korea, 3 Center for Happiness Studies, Seoul National University, Seoul, Republic of Korea, 4 Department of Family Medicine, College of Medicine, Kyung Hee University, Seoul, Republic of Korea

⊙ These authors contributed equally to this work.
* bumjo.oh@gmail.com

**Data Availability Statement:** All relevant data are within the paper and its Supporting information files.

## Abstract

### Introduction

Young adults receive health screenings at lower rates than other age groups, and it may be difficult to detect diseases in the early stages for this group. We examined differences in health status relative to smoking in a young age group using the results of health screenings conducted in engaged and newly married couples in a cross-sectional database.

### Methods

The participants in this study were 808 young adults who visited a municipal hospital health screening center from July 2017 to March 2019. They completed a self-administered questionnaire, and physical measurements and a blood test were taken. They were classified into non-cigarette smokers, past cigarette smokers, and current cigarette smokers according to smoking behavior. In this study, we compared metabolic syndrome, the main components of which include obesity, high blood pressure, high blood triglycerides, low levels of HDL cholesterol and insulin resistance, with smoking behavior.

### Results

The mean age of the participants was 30.9±3.3 years (males 32.0±3.2, females 29.8±3.1), and 13.9% were current cigarette smokers (males 22.8%, females 5.1%). The proportion of men in their 30s was 76.6% for male group and 50.0% for female group, indicating that the male group had a relatively higher proportion of older and current smokers. Significant differences were found in age, sex, blood pressure, metabolic abnormalities, and drinking status according to smoking status. Cigarette smokers had a 2.4-fold greater risk of metabolic syndrome (95% confidence interval [CI], 1.43–3.96) than non-cigarette smokers; in particular,

**Funding:** The authors received no specific funding for this work.

**Competing interests:** The authors have declared that no competing interests exist.

they had a 2.6-fold (95% CI, 1.44–4.55) greater risk of hypertriglyceridemia and a three-fold (95% CI, 1.45–6.35) greater risk of low HDL cholesterol.

## Conclusions

In comparison with non-single, young and generally healthy city dwellers, the risk of metabolic syndrome was significantly higher in smokers than in non-smokers, and in particular, it was confirmed that the risk of hypertriglyceridemia and low HDL cholesterolemia was higher. Smoking cessation is necessary, even for the young, because smoking may cause changes in blood lipids even if the smoking duration is short.

## Introduction

Recent demographic changes that have been occurring in South Korea include falling fertility and an aging population [1]. In particular, one of the noticeable changes that has occurred is the number of marriages and the typical age of first marriage. The annual number of marriages in South Korea increased over a long period, from 295.1 thousand in 1970 to 419.7 thousand in 1992, but decreased slightly afterwards to 332.0 thousand in 2000, 314.3 thousand in 2005, and then 326.1 thousand in 2010 [1]. As the number of marriages has been decreasing, the average age at first marriage has been increasing. A survey of the social and economic situation of young people in 2016 reported that 56.0% of respondents said that marriage was necessary, and the average age of their expected marriage was 30.1 years [2]. However, marriage and divorce statistics from the National Statistical Office in 2018 show an average age at first marriage of 33.2 years for men and 30.4 for women and an increase of 0.2 years for both men and women; these values were 1.8 years higher for men and 2.2 years higher for women than their equivalents 10 years ago [3]. The increase in marriage age is complexly connected with problems in education, including the extension of the time necessary to complete higher education, with labor market problems, such as a general lack of positions and lack of good quality ones in particular, with changes in norms and thinking about marriage, and with changes in the marriage market, including the pool of potential male and female partners [1]. Several analyses have indicated that this trend will not change.

Because the population group aged 20–30 years includes the current average age of marriage and is largely healthy, they often avoid proper health assessments. However, as marriage age increases, poor health in members of this age group can cause important health and sociological problems. Subjective health conditions were highest among those in their 20s (35.3%) followed by those in their 30s (32.9%) (National Health and Nutrition Survey, 2017) [4]. As subjective health conditions drop to 20% for those in in their 40s and older, it is highly likely that those in their 20s and 30s do not understand their health as well as other age groups. In particular, young men in their 20s and 30s had health checkup rates of 36.5–55.4%, lower than those for other age groups, making it difficult to diagnose hypertension, diabetes mellitus, or dyslipidemia early [4]. The rate of unmet medical care, which measures necessary medical services that are left unperformed, was also highest among men in their 30s, at 11.5%, and third highest among men in their 20s. The main reasons for unmet medical care included lack of time, symptoms that are not burdensome, and economic reasons [5]. Smoking rates were the second highest among men in their 30s, at 25.5%, and third highest among men in their 20s, at 24.4%. Monthly drinking rates were the highest among men in their 20s, at 50.6%, and second

highest among men in their 30s, at 42.6% [4], implying that men in their 20s and 30s do not take proper care of their health and tend to lead unhealthy lifestyles.

Metabolic syndrome refers to the co-occurrence of abdominal obesity, dyslipidemia, elevated blood pressure, and impaired fasting glucose. When metabolic syndrome is neglected, the risk of cardio-cerebrovascular disease, diabetes mellitus, and cancer increases.[6–8] The prevalence of metabolic syndrome is increasing in South Korea [9], but there have been no independent surveys of metabolic syndrome among persons in their 20s and 30s, who often do not properly assess their health.

In younger age groups, smoking habits may not be seriously taken because the duration of the habit is often not long, and many believe that they will have many chances to quit at a later time. If it can be confirmed that smoking among various lifestyles at a healthy young age is related to a chronic disease state represented by metabolic syndrome, it is expected that this will serve as a basis for recommending smoking cessation to them. Therefore, this study tried to investigate this association by analyzing the health checkup results of engaged and newlywed persons who visited a health examination center at a general hospital in Seoul from July 2017 to March 2019.

## Methods

### Study population and data collection

The study population was 808 people who visited the health promotion center of a municipal hospital from July 3, 2017, to March 31, 2019, including of 288 in 2017, 480 in 2018, and 40 in 2019, with no overlap. This population consisted of couples engaged or married less than one year, proved with documents such as wedding invitations, wedding hall use agreements, and resident registration certificates. The screening items were a self-administered questionnaire, anthropometry, blood tests, urine tests, stool tests, electrocardiograms, pulmonary function tests, image examination, and gastroscopy. Part of the cost of the examination was borne by the subjects, and the remainder was covered by a sponsorship fund and contributions.

### Ethics approval and consent to participate

This study was approved by the Institutional Review Board of SMG-SNU Boramae Medical Center (registration number: 30-2018-101). Written informed consent confirming voluntary participation was obtained from the participants. All individually identifying records were anonymized prior to analysis.

### Study method

**1) Anthropometry and health-related behavior survey.** Height and weight were measured with an automatic height–weight measuring instrument (same for all subjects). The subjects were fasting, wearing their examination clothes, and wearing no shoes. The measured height (cm) and weight (kg) values were recorded to one decimal place. The body mass index was calculated as weight (kg)/height (m$^2$). In the WHO Asian-Pacific criteria, body mass index values smaller than 18.5 kg/m$^2$ are underweight, 18.5 kg/m$^2$–23 kg/m$^2$ are normal weight, 23 kg/m$^2$–25 kg/ m$^2$ are overweight, and greater than 25 kg/m$^2$ are obese [10]. Waist circumferences were measured in the upright posture at the end of normal expiration at the midpoint between the lower border of the rib cage and the top of the iliac crest, following the WHO recommendations. Blood pressure was measured twice using an automatic blood pressure gauge after the subjects had relaxed for at least 10 minutes while they were sitting in a chair, leaning against the back of the chair, their feet placed on the floor without their legs crossed, and with the height of the upper arm adjusted to be equal to the height of the heart.

Health-related behaviors were identified using a self-administered questionnaire on smoking and drinking. The subjects were classified into non-cigarette smokers who had not smoked at least 100 cigarettes in their lifetime and were not currently smoking, previous cigarette smokers who had smoked at least 100 cigarettes in their lifetime but were not currently smoking, and current cigarette smokers who had smoked at least 100 cigarettes in their lifetime and were currently smoking. The kinds of alcoholic beverages drunk were collected. All subjects who reported that they did not drink alcohol at all were considered non-drinkers, and those who wrote that they drank one or more times were considered drinkers.

**2) Blood tests.**   Blood tests were conducted on serum separated from blood collected from the brachial vein after a state of fasting for at least 12 hours, and fasting blood glucose, glycated hemoglobin, total cholesterol, triglyceride, high-density lipoprotein (HDL) cholesterol, and low-density lipoprotein (LDL) cholesterol were measured using a Chemistry Analyzer (Roche-Hitachi).

**3) Definition of metabolic syndrome.**   The National Cholesterol Education Program/Adult Treatment Panel III defined metabolic syndrome as when at least three of five the following components appear: abdominal obesity, impaired fasting glucose, hypertriglyceridemia, low HDL cholesterolemia, and high blood pressure [11]. Abdominal obesity was assessed using a waist circumference of >90 cm for men and >85 cm for women. Impaired fasting glucose was defined as ≥100 mg/dL or use of an oral hypoglycemic agent or insulin. Hypertriglyceridemia was defined as ≥150 mg/dL or receipt of lipid-lowering drug therapy. Low HDL cholesterolemia was defined as an HDL level of <40 mg/dL for men and <50 mg/dL for women or receipt of lipid-lowering drug therapy. High blood pressure was defined as systolic blood pressure of ≥130 mmHg, or diastolic blood pressure of ≥85 mmHg, or taking a blood pressure medication.

**4) Statistical analysis.**   The general characteristics of the study subjects such as age, sex, blood pressure, individual metabolic abnormalities, and drinking status were examined using frequency analyses and compared using chi-square tests among the study subjects, who were classified into non-cigarette smokers, previous cigarette smokers, and current cigarette smokers [11]. In addition, after adjusting for age, sex, drinking, and underlying disease using multivariable logistic regression analyses, the effects of smoking state on metabolic syndrome and individual metabolic abnormalities were assessed. The survey data were compiled using Microsoft Excel and STATA ver. 15.0 was used for statistical analyses. Significance was assessed at p < 0.05.

## Results

### Characteristics of study subjects

The total number of subjects was 808, and the mean age was 30.88 ± 3.29 years. Among these, 398 were men, and 410 were women. Among the subjects, 72.52% were non-cigarette smokers, 13.61% were previous cigarette smokers, and 13.86% were current cigarette smokers; 67.20% were current drinkers. Metabolic syndrome was found in 17.45% of subjects; in 31.91% of men and 4.63% of women. The ratio of those with underlying diseases was only 2.60% of all subjects (Table 1).

### Comparison of characteristics of study subjects according to smoking status

Table 2 provides a comparison of general characteristics, such as age, sex, blood pressure, individual metabolic abnormalities, and drinking status among smoking status groups, which showed significant differences in all characteristics except underlying diseases (Table 2).

**Table 1. Descriptive characteristics of study participants.**

| Variables | Total | Men, N (%) | Women, N (%) |
|---|---|---|---|
| **Number of people** | | | |
| Proportion, N (%) | 808 | 398 (49.26) | 410 (50.74) |
| **Age, years** | | | |
| Mean (±SD) | 30.88 (3.29) | 31.99(3.15) | 29.80 (3.06) |
| 20–29 years old, N (%) | 298 (36.88) | 93 (23.37) | 205 (50.00) |
| 30–39 years old, N (%) | 510 (63.12) | 305 (76.63) | 205 (50.00) |
| **Blood pressure** | | | |
| Mean SBP (±SD) | 113.64 (13.43) | 121.05 (12.60) | 106.45(9.83) |
| Mean DBP (±SD) | 77.96 (9.80) | 81.91 (10.07) | 74.13 (7.82) |
| Normal, N (%) | 614 (75.99) | 245 (61.56) | 369 (90.00) |
| Increasing Blood Pressure, N (%) | 194 (24.01) | 153 (38.44) | 41 (10.00) |
| **Fasting blood glucose** | | | |
| Mean (±SD) | 86.93 (10.70) | 89.95 (12.09) | 84.01 (8.16) |
| Normal, N (%) | 759 (93.94) | 353 (88.69) | 406 (99.02) |
| Increasing Fasting blood glucose, N (%) | 49 (6.06) | 45 (11.31) | 4 (0.98) |
| **Abdominal obesity**[a] | | | |
| Mean (± SD) | 81.04 (10.78) | 88.33 (8.75) | 73.98 (7.31) |
| Normal weight, N (%) | 613 (75.87) | 241 (60.55) | 372 (90.73) |
| Obesity, N (%) | 195 (24.13) | 157 (39.45) | 38 (9.27) |
| **General obesity**[b] | | | |
| Mean (±SD) | 22.75 (3.61) | 24.89 (3.27) | 20.68 (2.56) |
| Normal weight, N (%) | 612 (75.74) | 221 (55.53) | 391 (95.37) |
| Obesity, N (%) | 196 (24.26) | 177 (44.47) | 19 (4.63) |
| **Metabolic syndrome**[c] | | | |
| Normal, N (%) | 667 (82.55) | 271 (68.09) | 396 (96.59) |
| Metabolic syndrome, N (%) | 141 (17.45) | 127 (31.91) | 14 (3.41) |
| **Smoking** | | | |
| Non-cigarette smoker | 586 (72.52) | 212 (53.27) | 374 (91.22) |
| Past cigarette smoker | 110 (13.61) | 95 (23.87) | 15 (3.66) |
| Current cigarette smoker | 112 (13.86) | 91 (22.86) | 21 (5.12) |
| **Alcohol use** | | | |
| Non-drinker | 262 (32.43) | 96 (24.12) | 166 (40.49) |
| Drinker | 543 (67.20) | 300 (75.38) | 243 (59.27) |
| **Presence of comorbidity**[d] | | | |
| No | 787 (97.40) | 380 (95.48) | 407 (99.27) |
| Yes | 21 (2.60) | 18 (4.52) | 3 (0.73) |

Values are presented as count number (weighted %).

[a]Abdominal obesity (Korean criteria): waist circumference 90 cm in men and $\geq$ 85 cm in women.

[b]General obesity (Korean criteria): body mass index (BMI) $\geq$ 25 kg m2.

[c]Metabolic syndrome (Korean criteria): person has three or more of the following measurements: 1. Abdominal obesity (waist circumference 90 cm in men, 85 in women), 2. Triglyceride level $\geq$ 150 mg/dL, 3. HDL cholesterol $<$ 40 mg/dL in men or $<$ 50 mg/dL in women or on dyslipidemia medication, 4. Systolic blood pressure (top number) $\geq$ 130 mm Hg or diastolic blood pressure (bottom number) $\geq$ 85 mmHg or on hypertension medication, 5. Fasting glucose $\geq$ 100 mg/dL or on diabetes medication.

[d]Comorbidity related to metabolic syndrome.

**Table 2. Comparison of subject characteristics according to smoking status.**

| Variables | Non-cigarette smoker | Past cigarette smoker | Current cigarette smoker | P-value[a] |
|---|---|---|---|---|
| **Number of people** | 586 (72.52) | 110 (13.61) | 112 (13.86) | |
| N (%) | | | | |
| **Age, years** | | | | <0.005 |
| 20–29 years | 246 (41.98) | 29 (26.36) | 23 (20.54) | |
| 30–39 years | 340 (58.02) | 81 (73.64) | 89 (79.46) | |
| **Sex** | | | | <0.005 |
| Men | 212 (36.18) | 95 (86.36) | 91 (81.25) | |
| Women | 374 (63.82) | 15 (13.64) | 21 (18.75) | |
| **Blood pressure** | | | | <0.005 |
| Normal, N (%) | 473 (80.72) | 70 (63.64) | 71 (63.39) | |
| Increasing blood pressure, N (%) | 113 (19.28) | 40 (36.36) | 41 (36.61) | |
| **Fasting blood glucose** | | | | 0.005 |
| Normal | 560 (95.56) | 100 (90.91) | 99 (88.39) | |
| Increasing fasting blood glucose | 26 (4.44) | 10 (9.09) | 13 (11.61) | |
| **Triglyceride** | | | | <0.005 |
| Normal | 541 (92.32) | 91 (82.73) | 75 (66.96) | |
| Increasing triglycerides | 45 (7.68) | 19 (17.27) | 37 (33.04) | |
| **High Density Lipoprotein** | | | | <0.005 |
| Normal | 550 (93.86) | 99 (90.00) | 94 (83.93) | |
| Decreasing HDL | 36 (6.14) | 11 (10.00) | 18 (16.07) | |
| **Alcohol use** | | | | <0.005 |
| Non-drinker | 223 (38.05) | 18 (16.36) | 21 (18.75) | |
| Drinker | 361 (61.60) | 92 (83.64) | 90 (80.36) | |
| **Presence of comorbidity[d]** | | | | 0.08 |
| No | 575 (98.12) | 106 (96.36) | 106 (94.64) | |
| Yes | 11 (1.88) | 4 (3.64) | 6 (5.36) | |
| **Abdominal obesity[a]** | | | | <0.005 |
| Normal weight | 477 (81.40) | 69 (62.73) | 67 (59.82) | |
| Obesity | 109 (18.60) | 41 (37.27) | 45 (40.18) | |
| **General obesity[b]** | | | | <0.005 |
| Normal weight | 480 (81.91) | 66 (60.00) | 66 (58.93) | |
| Obesity | 106 (18.09) | 44 (40.00) | 46 (41.07) | |
| **Metabolic syndrome[c]** | | | | <0.005 |
| Normal | 667 (82.55) | 271 (68.09) | 396 (96.59) | |
| Metabolic syndrome | 141 (17.45) | 127 (31.91) | 14 (3.41) | |

[a]Analyzed by chi-square test.

[a]Abdominal obesity (Korean criteria): waist circumference 90 cm in men and $\geq$ 85 cm in women.

[b]General obesity (Korean criteria): body mass index (BMI) $\geq$ 25 kg m2.

[c]Metabolic syndrome (Korean criteria): person has three or more of the following measurements: 1. Abdominal obesity (waist circumference 90 cm in men, 85 in women), 2. Triglyceride level $\geq$ 150 mg/dL, 3. HDL cholesterol $<$ 40 mg/dL in men or $<$ 50 mg/dL in women or on dyslipidemia medication, 4. Systolic blood pressure (top number) $\geq$ 130 mm Hg or diastolic blood pressure (bottom number) $\geq$ 85 mmHg or on hypertension medication, 5. Fasting glucose $\geq$ 100 mg/dL or on diabetes medication.

[d]Comorbidity related to metabolic syndrome.

## Prevalence rates of metabolic syndrome and individual metabolic abnormalities according to smoking status

Table 3 shows the associations between smoking, obesity, metabolic syndrome, and individual components of metabolic syndrome after covariate adjustment. Cigarette smokers had a 2.38

**Table 3. Crude and adjusted odds ratios (and 95% confidence intervals) from logistic regression analyses identifying associations between smoking status and components of metabolic syndrome.**

| Components | Non-smoker | Past smoker | Current smoker |
|---|---|---|---|
| Proportion, N (%) | 586 (72.52) | 110 (13.61) | 112 (13.86) |
| General obesity[a] | 106 (18.09) | 44 (40.00) | 46 (41.07) |
| Crude OR (95% CI) | 1 | **3.02** (1.95–4.67) | **3.22** (2.10–4.95) |
| Adjusted OR[b] (95% CI) | 1 | 1.10 (0.68–1.79) | 1.23 (0.75–2.01) |
| Metabolic syndrome[c] | 30 (5.12) | 15 (13.64) | 21 (18.75) |
| Crude OR (95% CI) | 1 | **2.20** (1.32–3.67) | **5.34** (3.40–8.37) |
| Adjusted OR[d] (95% CI) | 1 | 0.88 (0.50–1.53) | **2.38** (1.43–3.96) |
| Abdominal obesity[e] | 109 (18.60) | 41 (37.27) | 45 (40.18) |
| Crude OR (95% CI) | 1 | **2.60** (1.68–4.03) | **2.90** (1.88–4.45) |
| Adjusted OR[f] (95% CI) | 1 | 1.24 (0.76–2.01) | 1.43 (0.89–2.31) |
| Increased blood pressure | 113 (19.28) | 40 (36.36) | 41 (36.61) |
| Crude OR (95% CI) | 1 | **2.39** (1.54–3.71) | **2.42** (1.56–3.74) |
| Adjusted OR[g] (95% CI) | 1 | 1.13 (0.68–1.88) | 0.94 (0.56–1.59) |
| Increased fasting blood glucose | 26 (4.44) | 10 (9.09) | 13 (11.61) |
| Crude OR (95% CI) | 1 | **2.15** (1.00–4.60) | **2.83** (1.41–5.69) |
| Adjusted OR[h] (95% CI) | 1 | 0.94 (0.42–2.12) | 1.03 (0.47–2.25) |
| Increased triglycerides | 45 (7.68) | 19 (17.27) | 37 (33.04) |
| Crude OR (95% CI) | 1 | **2.51** (1.40–4.48) | **5.93** (3.61–9.75) |
| Adjusted OR[i] (95% CI) | 1 | 1.04 (0.55–1.98) | **2.56** (1.44–4.55) |
| Decreased high-density lipoprotein | 36 (6.14) | 11 (10.00) | 18 (16.07) |
| Crude OR (95% CI) | 1 | 1.70 (0.84–3.45) | **2.93** (1.59–5.37) |
| Adjusted OR[j] (95% CI) | 1 | 2.26 (0.98–5.20) | **3.03** (1.45–6.35) |

OR, odds ratio; CI, confidence intervals. The numbers in bold indicate statistically significant differences in (p < .05) odds for smoking status in a given group as compared to the reference (non-smoker) group.

[a]General obesity (Korean criteria): body mass index (BMI) $\geq$ 25 kg m$^2$.

[b]Adjusted for age, sex, drinking, comorbidity.

[c]Metabolic syndrome (Korean criteria): person has three or more of the following measurements: 1. Abdominal obesity (waist circumference 90 cm in men, 85 in women), 2. Triglyceride level $\geq$ 150 mg/dL, 3. HDL cholesterol < 40 mg/dL in men or < 50 mg/dL in women or on dyslipidemia medication, 4. Systolic blood pressure (top number) $\geq$ 130 mm Hg or diastolic blood pressure (bottom number) $\geq$ 85 mmHg or on hypertension medication, 5. Fasting glucose $\geq$ 100 mg/dL or on diabetes medication.

[d]Adjusted for age, sex, drinking, comorbidity.

[e]Abdominal obesity (Korean criteria): waist circumference >90 cm in men, >85 in women.

[f]Adjusted for age, sex, drinking, comorbidity.

[g]Adjusted for age, sex, drinking, comorbidity, waist circumference, triglycerides, high density lipoprotein, Fasting blood glucose.

[h]Adjusted for age, sex, drinking, comorbidity, waist circumference, triglycerides, high density lipoprotein, blood pressure.

[i]Adjusted for age, sex, drinking, comorbidity, waist circumference, high density lipoprotein, blood pressure, Fasting blood glucose.

[j]Adjusted for age, sex, drinking, comorbidity, waist circumference, triglycerides, blood pressure, Fasting blood glucose.

times higher risk for metabolic syndrome than non-cigarette smokers, 2.56 times higher risk for hypertriglyceridemia, and 3.03 times higher risk for low HDL cholesterolemia.

## Discussion

In this study, the authors found that smoking was significantly associated with metabolic syndrome in urban living and healthy young populations. Compared to nonsmokers, smokers had a 2.4 times higher risk of metabolic syndrome. Moreover, the current findings support that smoking increases blood triglyceride levels and lowers HDL cholesterol levels, indicating that the changes in blood lipid levels caused by smoking may play an important role in the association between smoking and metabolic syndrome [12].

Smoking induces an increase in body insulin-antagonistic hormones, such as cortisol, catecholamine, and growth hormone, and it induces increases in lipolysis [13]. A previous finding indicated that this increased triglyceride levels [14]. In addition, nicotine levels in the body promote fat degradation. The free fatty acids increased through these pathways damage the beta cells of the pancreas, which is thought to cause impaired fasting glucose [15]. It was found in the literature that by synthesizing these mechanisms, metabolic syndrome-related factors are significantly identified in diabetic smokers [16].

In young adults in their 20s and 30s, the duration or amount of smoking is likely to be relatively low compared to older age groups, but among those who were cigarette smokers, the risk of dyslipidemia was significantly higher than among non-cigarette smokers. In addition, although this was not statistically significant, previous cigarette smokers had a higher risk of dyslipidemia than non-cigarette smokers. Combining the previous findings, we can infer that current or past smoking is associated with dyslipidemia, separate from the period of smoking. In order to confirm this, an additional comparison can be made by classifying light or heavy smokers according to the amount and duration of smoking, but we divided them into three groups: current smokers, non-smokers, and former smokers. This was because the definition of light smoker was not clear and the number of subjects was small, so if the category was further divided by the new definition, statistical power could be reduced. However, considering that there is a possibility of a dose-response relationship between the amount of smoking and metabolic syndrome, it was thought that large-scale data analysis would be necessary regarding this point in the future.

The risk of metabolic syndrome was statistically significantly higher among cigarette smokers than non-cigarette smokers. Previous studies indicate that smoking is significantly associated with metabolic syndrome [17–19]. Moreover, among the components of metabolic syndrome, abdominal obesity, elevated blood pressure, and impaired fasting glucose were found to have a higher prevalence in the smoking group [18]. In this study, cigarette smokers and previous cigarette smokers showed a higher risk for abdominal obesity and elevated blood pressure than non-cigarette smokers, although the differences were not statistically significant. Given the results obtained with the 20–30 age group, who are the subjects of this study, and the results of previous studies, it is thought that smoking increases the risk of metabolic syndrome regardless of age at the beginning of smoking or the duration of smoking. However, compared to the non-smoker group, the previous cigarette smokers group was not significantly different not only in metabolic syndrome and its components, but also in general obesity. This suggests that quitting smoking at a young age may lower the risk of metabolic syndrome compared to cigarette smokers.

Previous studies report that more cigarette smokers have poor lifestyle habits, including drinking and inactivity, in addition to smoking, and these habits increase the risk of metabolic disease [20]. In particular, it is well known that alcohol consumption is a major cause of

metabolic syndrome. However, a possible limitation of this study is that when the questionnaires were collected, the authors were unable to accurately determine the type and amount of alcohol that subjects drank. For the convenience of the research, a drinker is defined as "drinking more than once a week," so the proportion of drinkers is high. Therefore, we acknowledge that more meaningful results could have been confirmed if the subjects were classified into moderate/risk drinking according to the amount of alcohol consumed and the relationship with metabolic syndrome risk factors had been further analyzed. Although this study's findings cannot be compared with previous work because the amount of drinking and activity was not evaluated, this tendency to have poor habits is believed to appear in this age group. This study suggests that even relatively young age groups should pay attention to their lifestyle habits. After the above findings were combined, it was seen that smoking cessation could be an important improvement in lifestyle habits that lowers the prevalence of dyslipidemia and metabolic syndrome.

The cross-sectional studies we conducted have the following general limitations. First, because it was a cross-sectional study, estimating the cause-effect relationship between smoking and metabolic syndrome was complicated. Second, this study cannot be used to represent the entire population in this age group because it was conducted only among those who sought a health checkup at a municipal hospital within a specific period of time. Third, because the amount of drinking was measured in a self-administered questionnaire, it was challenging to evaluate drinking accurately. This may have produced errors during statistical analysis.

Nevertheless, to the best of the our knowledge, this may be a very rare study of smoking and metabolic syndrome in a young, healthy adult population under 40 years of age. Other studies using health screening data have mainly concentrated on older age groups, and no study has been conducted with relatively young age groups. A significant strength of this study is its identification of significant results, although the subjects were young adults with short smoking durations. It is suggested that subsequent studies with more sophisticated designs be performed to overcome some of the aforementioned limitations. In addition, based on the current results, it is expected that this will serve as a basis for recommending quitting smoking among young people with short smoking periods and no other diseases.

## Supporting information

**S1 Dataset.**
(DTA)

## Author Contributions

**Conceptualization:** Sun Woo Kim, Ho Jun Kim, Hobeom Lee, Sunyoung Kim, Jong Seung Kim, Bumjo Oh.

**Data curation:** Hobeom Lee, Sung-Ha Lee, Jong Seung Kim.

**Formal analysis:** Sun Woo Kim, Ho Jun Kim, Kyungha Min, Sung-Ha Lee, Sunyoung Kim.

**Methodology:** Kyungha Min.

**Project administration:** Bumjo Oh.

**Supervision:** Sunyoung Kim, Jong Seung Kim, Bumjo Oh.

**Writing – original draft:** Sun Woo Kim.

**Writing – review & editing:** Ho Jun Kim, Kyungha Min, Hobeom Lee, Sung-Ha Lee, Jong Seung Kim.

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
