## [Decision Letter · Decision Letter 0]

6 May 2021

PONE-D-20-15624

Relationship between Smoking and Metabolic Syndrome in Engaged and Newly Married Couples Adults in Their 20s and 30s

PLOS ONE

Dear Dr. Oh,

Thank you for submitting your manuscript to PLOS ONE. After careful consideration, we feel that it has merit but does not fully meet PLOS ONE’s publication criteria as it currently stands. Therefore, we invite you to submit a revised version of the manuscript that addresses the points raised during the review process.

We look forward to receiving your revised manuscript.

Kind regards,

Liou Y. Sun

Academic Editor

PLOS ONE

Journal Requirements:

 [No].

Reviewers' comments:

Reviewer's Responses to Questions

**Comments to the Author**

1. Is the manuscript technically sound, and do the data support the conclusions?

Reviewer #1: Yes

Reviewer #2: Partly

2. Has the statistical analysis been performed appropriately and rigorously? 

Reviewer #1: Yes

Reviewer #2: Yes

3. Have the authors made all data underlying the findings in their manuscript fully available?

Reviewer #1: Yes

Reviewer #2: Yes

4. Is the manuscript presented in an intelligible fashion and written in standard English?

Reviewer #1: Yes

Reviewer #2: Yes

5. Review Comments to the Author

Reviewer #1: This is an important area of research to explore how smoking affects the cardiometabolic parameters in people in their 20s and 30s. This paper can be published with some minor corrections. The paper needs overall editing for grammar and punctuation.

Reviewer #2: Obesity and smoking are important causes of morbidity and mortality worldwide. Smoking reduces insulin sensitivity, induces insulin resistance, and enhances cardiovascular risk factors, such as elevated plasma triglycerides, reduced high-density lipoprotein–cholesterol, and hyperglycemia. Smoking is associated with metabolic abnormalities and increases the risk of metabolic syndrome.

In this study, the authors evaluated the relationship between smoking and metabolic syndrome in 808 young adults and found that the odds ratio versus nonsmokers for metabolic syndrome, hypertriglyceridemia, and low HDL cholesterolemia was significantly higher in smokers. This study provides a positive association between smoking and metabolic syndrome in young adults.

Some questions need to be addressed. Whether and how drinking affects metabolic syndrome in young adults either in the smoking group or non-smoking group. Was any difference of metabolic syndrome analyzed in the light smokers and heavy smokers, compared to the non-smoking group?

6. PLOS authors have the option to publish the peer review history of their article (what does this mean?). If published, this will include your full peer review and any attached files.

Reviewer #1: **Yes: **Debasish Kar

Reviewer #2: No

---

## [Author Response · Author response to Decision Letter 0]

21 Jul 2021

Dear Editor,

We like to thank you and the reviewers for the very helpful advices.

Please find our revision and answers to the open points enclosed. 

Best regards,

ABSTRACT

1) Title – I am not quite sure why the authors have chosen ‘marriage’ as a determinant of metabolic syndrome? Marriage may not happen, and people can live together for decades. It does not affect their risk of developing metabolic syndrome. 

Answer: We agree with the reviewer's opinion. This was a kind of business, and the purpose was to recommend the health care of newlyweds, and I think that the title of this manuscript, written for research purposes, needs to be revised. So, we have changed it to:

The relationship between smoking cigarettes and metabolic syndrome: a cross-sectional study with non-single residents of Seoul under 40 years old

2) Metabolic syndrome was defined using Adult Treatment Panel III.- Please tell the reader what is meant by the adult treatment panel.

Answer: Thank you for your comment. The content was explained by quoting from the methodology, not the abstract.

3) Results – Please add standard deviation of age and male female split. This is going to help the reader to understand dispersion of data. Male female split will be helpful as you have mentioned the uptake of screening tests are comparatively lower in young males.

Answer: Thank you for your comment. Men and women were separated and SD was indicated.

4) Conclusion – Glycosylated Haemoglobin, smoking and metabolic syndrome are closely linked. Why the authors only commented on TG and HDL and not other components of the metabolic syndrome?

Answer: In this study, we looked at the relationship between smoking and metabolic syndrome, and further looked at the relationship between the components that define metabolic syndrome. Through this, the Conclusion emphasized and mentioned this part with the result that TG and HDL had a significant relationship with smoking.

INTRODUCTION

1) The annual number of marriages in South Korea showed increases over a long period – correct grammar. 

Answer: We have corrected the sentence as you pointed out.

2) “Metabolic syndrome is the composite occurrence of abdominal obesity, dyslipidemia, elevated blood pressure, and impaired fasting glucose” – what about BMI and waist circumference?

Answer: Thank you for your comment. The relationship between obesity index and metabolic syndrome risk factors has been suggested in various ways, and the authors selected one of them according to the criteria of Adult Treatment Panel III. This is because BMI and waist circumference are both used as indicators of obesity, but if they are selected at the same time, covariance may play a role.

3) “Therefore, this study is intended to conduct cross-sectional studies based on the results of health screenings of engaged and newly married couples who visited the health examination center of a general hospital in Seoul from July 2017 to March 2019 in order to analyze differences in the health status, various clinical tests, and prevalence between smoking and non-smoking among existing living habits in relatively young age groups” – too long sentence not sure what the authors are trying to say. Please rewrite the sentence.

Answer: We agree with the reviewer's opinion, and have corrected the sentence as pointed out.

STUDY METHOD

1) Statistical analysis – “In addition, after compensating for age, sex, drinking, ……” adjusting would be a better word than compensating.

Answer: We agree with the reviewer's opinion, and changed the word as you instructed.

2) Table 3 – Can the author clarify what is meant by crude OR and adjusted OR, please?

Answer: As you have pointed out, we added a description in Table 3.

DISCUSSION

1) Why the authors decided only to comment of lipid profiles rather than the whole spectrum of components of metabolic syndrome?

Answer: As mentioned in the Conclusion, this study confirmed that the prevalence of metabolic syndrome was higher in the current smoker group than in the non-smoker group. In particular, it was confirmed that the adjusted OR significantly increased in TG and HDL-C (high-density lipoprotein) among metabolic components. In this regard, the discussion was focused on the relevant risk factors. The authors emphasize this intention by amending the first sentence of the first paragraph of the Discussion.

2) Relationship between smoking and the components of metabolic syndrome is well established - https://pubmed.ncbi.nlm.nih.gov/27881170/

Answer: Thank you for your comment. We cited and mentioned it in the last sentence of the second paragraph of the Discussion after reviewing the contents of the review article.

3) The authors could inform the reader how this relationship is different in young people in their 20s and 30s. 

Answer: Thank you for your comment. The aforementioned papers are included as references. In addition, compared with previous studies, a significant increase in TG and a significant decrease in HDL were characteristic in smokers who were younger subjects under the age of 40, and this was mentioned in the Discussion section.

5. Review Comments to the Author

Reviewer #1: This is an important area of research to explore how smoking affects the cardiometabolic parameters in people in their 20s and 30s. This paper can be published with some minor corrections. The paper needs overall editing for grammar and punctuation.

Answer: Thank you for your comment. We have completed language editing once again for the revised manuscript.

Reviewer #2: Obesity and smoking are important causes of morbidity and mortality worldwide. Smoking reduces insulin sensitivity, induces insulin resistance, and enhances cardiovascular risk factors, such as elevated plasma triglycerides, reduced high-density lipoprotein–cholesterol, and hyperglycemia. Smoking is associated with metabolic abnormalities and increases the risk of metabolic syndrome.

In this study, the authors evaluated the relationship between smoking and metabolic syndrome in 808 young adults and found that the odds ratio versus nonsmokers for metabolic syndrome, hypertriglyceridemia, and low HDL cholesterolemia was significantly higher in smokers. This study provides a positive association between smoking and metabolic syndrome in young adults.

Some questions need to be addressed. Whether and how drinking affects metabolic syndrome in young adults either in the smoking group or non-smoking group. Was any difference of metabolic syndrome analyzed in the light smokers and heavy smokers, compared to the non-smoking group?

Answer: Thank you for your comments, and we would like to respond to them.

First, the researchers divided the subjects into three groups, namely current smokers, non-smokers, and ex-smokers, according to the definition of ever-smoker, and analyzed them.

1) Since the number of subjects is not large, if the categories are further divided by the new definition, the statistical power may decrease.

2) The definition of light/heavy smoker is not clear.

3) The smoking rate of female subjects is very low. 

For these reasons, the authors judged that it would be difficult to identify a significant difference between groups when additional conditions were added for analysis. However, we thought that there is a possibility of a dose-response between the amount of smoking and metabolic syndrome, so this was additionally described in the Discussion section.

Second, we are well aware that alcohol consumption is also a major contributor to metabolic syndrome. However, it was later realized that the authors did not accurately determine the type and amount of alcohol the subjects drank when collecting the questionnaire. For convenience, since “if you drink more than once a week” is defined as a drinker, it is acknowledged that the proportion of drinkers is high. Therefore, the authors agreed that a more meaningful result would have been possible if the relationship with metabolic components was further checked by dividing it into appropriate/risk drinking according to the amount of alcohol consumed. Reflecting these details, an existing reference was added and this content was additionally described as a limitation in the Discussion section.

---

## [Decision Letter · Decision Letter 1]

4 Aug 2021

The relationship between smoking cigarettes and metabolic syndrome: A cross-sectional study with non-single residents of Seoul under 40 years old

PONE-D-20-15624R1

Dear Dr. Bumjo Oh,

We’re pleased to inform you that your manuscript has been judged scientifically suitable for publication and will be formally accepted for publication once it meets all outstanding technical requirements.

Kind regards,

Liou Y. Sun

Academic Editor

PLOS ONE

Additional Editor Comments (optional):

two reviewers agree that the authors have addressed all the concerns.

Reviewers' comments:

Reviewer's Responses to Questions

**Comments to the Author**

1. If the authors have adequately addressed your comments raised in a previous round of review and you feel that this manuscript is now acceptable for publication, you may indicate that here to bypass the “Comments to the Author” section, enter your conflict of interest statement in the “Confidential to Editor” section, and submit your "Accept" recommendation.

Reviewer #1: All comments have been addressed

Reviewer #2: All comments have been addressed

2. Is the manuscript technically sound, and do the data support the conclusions?

Reviewer #1: Yes

Reviewer #2: Yes

3. Has the statistical analysis been performed appropriately and rigorously? 

Reviewer #1: Yes

Reviewer #2: Yes

4. Have the authors made all data underlying the findings in their manuscript fully available?

Reviewer #1: Yes

Reviewer #2: Yes

5. Is the manuscript presented in an intelligible fashion and written in standard English?

Reviewer #1: Yes

Reviewer #2: Yes

6. Review Comments to the Author

Reviewer #1: Few comments made for the authors to pay attention to. Otherwise it is a publishable quality research.

Reviewer #2: The authors have provided a detailed and thorough response to the comments from the previous review and have addressed my major concerns with the updated manuscript. The manuscript now reads with greater focus and

clarity.

7. PLOS authors have the option to publish the peer review history of their article (what does this mean?). If published, this will include your full peer review and any attached files.

Reviewer #1: **Yes: **Dr Debasish Kar

Reviewer #2: No

---

## [Editor Report · Acceptance letter]

10 Aug 2021

PONE-D-20-15624R1 

The relationship between smoking cigarettes and metabolic syndrome: a cross-sectional study with non-single residents of Seoul under 40 years old 

Dear Dr. Oh:

I'm pleased to inform you that your manuscript has been deemed suitable for publication in PLOS ONE. Congratulations! Your manuscript is now with our production department. 

Kind regards, 

on behalf of

Dr. Liou Y. Sun 

Academic Editor

PLOS ONE